# Advax-SM™-Adjuvanted COBRA (H1/H3) Hemagglutinin Influenza Vaccines

**DOI:** 10.3390/vaccines12050455

**Published:** 2024-04-24

**Authors:** Pedro L. Sanchez, Greiciely Andre, Anna Antipov, Nikolai Petrovsky, Ted M. Ross

**Affiliations:** 1Center for Vaccines and Immunology, University of Georgia, Athens, GA 30602, USA; sanchep3@ccf.org; 2Department of Infectious Diseases, University of Georgia, Athens, GA 30602, USA; 3Florida Research and Innovation Center, Cleveland Clinic, Port Saint Lucie, FL 34987, USA; 4Vaxine Pty Ltd., Adelaide, SA 5046, Australia; greiciely.andre@vaxine.net (G.A.); anna.antipov@vaxine.net (A.A.); nikolai.petrovsky@vaxine.net (N.P.); 5Department of Infection Biology, Lerner Research Institute, Cleveland Clinic, Cleveland, OH 44195, USA

**Keywords:** influenza, vaccine, adjuvant, Advax-SM™, COBRA

## Abstract

Adjuvants enhance immune responses stimulated by vaccines. To date, many seasonal influenza vaccines are not formulated with an adjuvant. In the present study, the adjuvant Advax-SM™ was combined with next generation, broadly reactive influenza hemagglutinin (HA) vaccines that were designed using a computationally optimized broadly reactive antigen (COBRA) methodology. Advax-SM™ is a novel adjuvant comprising inulin polysaccharide and CpG55.2, a TLR9 agonist. COBRA HA vaccines were combined with Advax-SM™ or a comparator squalene emulsion (SE) adjuvant and administered to mice intramuscularly. Mice vaccinated with Advax-SM™ adjuvanted COBRA HA vaccines had increased serum levels of anti-influenza IgG and IgA, high hemagglutination inhibition activity against a panel of H1N1 and H3N2 influenza viruses, and increased anti-influenza antibody secreting cells isolated from spleens. COBRA HA plus Advax-SM™ immunized mice were protected against both morbidity and mortality following viral challenge and, at postmortem, had no detectable lung viral titers or lung inflammation. Overall, the Advax-SM™-adjuvanted COBRA HA formulation provided effective protection against drifted H1N1 and H3N2 influenza viruses.

## 1. Introduction

Influenza viruses are part of the family *Orthomyxoviridae* and cause annual respiratory infections that can lead to hospitalization and fatal outcomes [1,2]. Influenza viruses can undergo antigenic drift and shift [3,4], thus allowing these viruses to evade host immune responses directed against the viral glycoproteins hemagglutinin (HA) and neuraminidase (NA), which are located on the virion surfaces [5,6,7]. Designing effective vaccines against the ever-changing influenza A viruses (IAVs) is challenging. Each season, wild-type (WT) influenza strains are selected for vaccine formulation and are not necessarily matched to circulating influenza virus strains, thus compromising the effectiveness of the vaccine [8,9]. Most commercially-available influenza vaccines are based on split-inactivated virus, recombinant hemagglutinin, or live-attenuated influenza virus (LAIV) platforms that induce protective immune responses through various mechanisms, including increased serum-neutralizing antibodies in the case of the non-live platforms and enhance mucosal immunity in the case of LAIV [10,11]. Adjuvants are molecular entities that enhance immune responses when co-administered with vaccine antigens [12,13]. Currently, the squalene oil-in-water emulsion, MF59, is the only adjuvant included in an FDA-licensed influenza vaccine which is indicated for use in the elderly [14,15]. AddaVax (SE adjuvant) is a commercially available squalene oil-in-water adjuvant designed to replicate MF59 [16,17].

To rectify the need for next-generation influenza vaccines, our group has previously developed novel antigens, referred to as computationally optimized broadly reactive antigen (COBRA). This method was utilized for synthesizing rHA and rNA proteins, capable of stimulating antibodies that induce broad responses, for providing long lasting protection against influenza virus strains that have undergone antigenic drift [18]. COBRA HA and NA virus-like particle vaccines stimulate antibodies with protective HAI and NAI responses against influenza A and B viruses in several animal models, including rodents, ferrets, and monkeys following vaccination [8,19,20,21,22].

Adjuvants may be particularly useful to enhance vaccine immunogenicity in groups that are at higher risks, like children with untrained immune systems or the immunosenescence seen in the elderly [18,19,23]. In this study, the adjuvant formulation, Advax-SM™ (https://protect-us.mimecast.com/s/WtgVCrk5zWtwyoOmLT7a_P5?domain=vac.niaid.nih.gov (accessed on 25 February 2024)) in combination with recombinant COBRA HA antigens was compared to a SE adjuvant (AddaVax) to assess their relative ability to protect mice against influenza viruses.

Advax-SM™ is synthesized from a natural plant-derived polysaccharide, inulin, that has immunostimulatory properties when crystalized into its delta polymeric form (delta-inulin) and formulated with TLR9 agonist CpG oligonucleotides [24]. Like other classes of non-immunomodulatory particulates, delta-inulin stimulates the recruitment of immune cells to the site of injection for enhancing the activity of presenting exogenous vaccine antigens to APCs that in turn presents antigens to CD4+ and CD8+ T-cells [25]. TLR9 agonist CpG oligonucleotides directly induce the activation of dendritic cells and enhance differentiation of B cells into antibody-secreting plasma cells [26,27,28]. Advax-SM™ adjuvant increases protective antibody titers and induces strong B and T cell responses with a significant rise in IgM and IgG levels [25]. Moreover, vaccines formulated with Advax-SM™ elicited enhanced Th1 (IL-2, IFN-γ) and Th2 (IL-5, IL-6) cytokines [25] following either intramuscular or subcutaneous injections [25,29,30,31,32].

To enhance the immune responses elicited by COBRA HA vaccines, mice were vaccinated with COBRA HA proteins representing the H1 and H3 subtypes of influenza that were mixed with Advax-SM™ or SE adjuvant. Overall, the goal of this study was to assess how immune responses induced by vaccination with COBRA HA vaccines formulated with a novel particulate polysaccharide adjuvant containing CpG55.2 TLR9 agonist, Advax-SM™, compare to a SE adjuvant and how these responses might elicit the most protective immune responses against a diverse panel of H1N1 and H3N2 influenza viruses.

## 2. Materials and Methods

### 2.1. COBRA HA Designs and Protein Synthesis

COBRA HA antigens corresponding to H1N1 and H3N2 seasonal influenza viruses (IAVs) were developed using the next-generation COBRA methodology [18]. H1 COBRA HA, Y2, was originated by extracting full-length HA sequences pertaining to 6232 GISAID (https://gisaid.org (accessed on 24 February 2024)) wild-type influenza A(H1N1) viruses. H1N1 sequences of infections collected from human isolates ranging between 1 January 2014 to 31 December 2016 were used to download the HA residues 1–566 (Methionine as the first amino acid) from online databases (version 11.0.9+11 Geneious bioinformatics software, Biomatters, Ltd. Auckland, New Zealand) and organized in order of collection date [19].

H3 COBRA HA, NG2, was originated by extracting full length HA sequences pertaining to 22,144 GISAID (https://gisaid.org (accessed on 24 February 2024)) human wild-type influenza A(H3N2) viruses. H3N2 sequences of infections collected from human isolates ranging between 1 January 2016 to 31 December 2018 were used to download (version 11.0.9+11 Geneious bioinformatics software, Biomatters, Ltd. Auckland, New Zealand) the HA residues 1–566 (Methionine as the first amino acid) from an online database and organized in order of collection.

Soluble COBRA HAs were purified from cells transfected with plasmids containing a pcDNA3.1 with an incorporated truncated HA gene that was made by substituting the transmembrane domain with a T4 fold-on domain, an Avitag, and a 6× His-tag [33]. Concentrations of the soluble HA molecules were measured via the bicinchoninic acid assay (BCA). Both the full-length HA sequences used in this study and the multiple sequence alignment are indicated in Table 1 (for H1N1 HAs) and Table 2 (for H3N2 HAs).

### 2.2. Vaccination

DBA/2J female mice (n = 75; 6 to 8 weeks in age) were purchased from Jackson Laboratory (Bar Harbor, ME, USA). All of the mice were contained in microisolator cages and had access to drinking water and food. USDA guidelines for laboratory animals were followed for caring for the mice and all of the procedures performed on the mice were reviewed and approved by the University of Georgia Institutional Animal Care and Use Committee (IACUC) (no. A2020 03-007-Y2-A7). Prior to starting the study, all of the mice were randomized and separated into different experimental groups with an n = 18 per group (Figure 1a). The remaining three naïve mice were used as comparers for lung pathology only, following infection. Prior to vaccination (day 0), serum samples were collected to verify that each mouse was seronegative, having no antibodies against the following viruses A/Texas/36/1991 (TX/91), A/Brisbane/59/2007 (Bris/07), A/California/07/2009 (Cal/09), A/Brisbane/02/2018 (Bris/18), and H3N2 viruses A/Texas/50/2012 (TX/12), A/Switzerland/9715293/2013 (SW/13), A/Hong Kong/4801/2014 (HK/14), A/Singapore/IFNIMH/2016 (Sing/16), A/Kansas/14/2017 (KS/17), A/Switzerland/8060/2017 (SW/17), A/Hong Kong/2671/2019 (HK/19), and A/South Australia/34/2019 (SA/19). Red blood cells were isolated from sera by centrifugation at a speed of ten thousand rpm for ten minutes and the sera samples were then stored at −20 °C ± 5 °C. Some of the mice were vaccinated intramuscularly (IM) with 3 μg of Y2 and 3 μg of NG2 COBRA HA antigens, and adjuvanted with 1 mg of Advax-SM™. Other mice were vaccinated with COBRA HA antigens as before and adjuvanted with SE adjuvant (AddaVax) at a 1:1 ratio. The mice pertaining to the control groups were vaccinated with no adjuvant or mock vaccinated with Advax-SM™ alone (Figure 1a,b). For boost vaccinations, all of the mice were intramuscularly vaccinated at the fourth and eighth weeks, as before. After the last boost, each mouse was bled on days 70 and 76 and the sera was processed via centrifugation and pooled, prior to storing at −20 °C ± 5 °C. For challenge (day 82), the mice were anesthetized and intranasally infected with 8 × 10^6^ PFU of A/Bris/18 (H1N1) influenza A virus (Figure 1a,b). Once the mice recovered from anesthesia, they were placed back into their cages and monitored for morbidity, mortality, and clinical signs (0–3; 0 = no clinical signs; 1 = weight loss 15- < 20%; 2 = dyspnea; 3 = weight loss > 20%/failure to respond to external stimulus/severe respiratory distress/neurological signs), for 14 days after. A mean clinical score of 3 was considered the endpoint and the animal was humanely euthanized via carbon dioxide asphyxiation followed by secondarily cervical dislocation. At day 85, vaccinated mice were euthanized as before and lungs were harvested from three mice in each group. The right lobes were snap-frozen on dry ice and stored at −80 °C for determining viral lung titers.

### 2.3. Viral Lung Titers

To determine viral lung titers, 1 × 10^6^ per 10 cm^2^ MDCK cells were added to tissue culture plates (Fisher Scientific, Pittsburgh, PA, USA) and placed in a static incubator for twenty-four hours at 37 °C + 5% CO_2_ until a monolayer of about ninety-five percent confluency was attained. Each mouse’s lungs were weighed and subjected to homogenizing in DMEM media containing 1 percent penicillin–streptomycin (P/S), in volumes of 10 times the weights of the lungs. Homogenized samples were subsequently centrifuged at one thousand five hundred rpm for ten minutes and serially diluted at 10-folds. For a positive control, Bris/18 was serial diluted as before. A total of 100 μL of the diluted homogenates were added to monolayers of MDCK cells to infect them for 60 min at RT, with fifteen-minute shaking interims. Some wells served as negative controls and 100 μL of DMEM P/S only was added to them. Following the 60 min incubation, the supernatants were removed from each well, laved one time with DMEM P/S media, and overlayed with 2 mL of a 1:1 solution of 1.6% agarose in 2X cMEM media + TPCK-Trypsin at 1 μg/mL. The plates were then placed in a static incubator for two to five days at 37 °C + 5% CO_2_. Upon confirmation of visible cytopathic effects, the agarose was detached and the monolayer of cells were fixed with ten percent formalin for ten minutes at room temperature. Subsequently, the formalin was discarded from each well and the monolayers were stained for ten to fifteen minutes at RT with one percent Crystal Violet (Fisher Science Education, Waltham, MA, USA). Lastly, following the incubation period, the Crystal Violet was discarded from each well and the monolayers were rinsed with water. The plates were allowed to air-dry and the plaque forming units (PFUs) were determined and calculated as PFU/g of tissue.

### 2.4. Histopathology

On day 85, lungs were harvested from three mice in each group. The left lobes of each lung were inflated with 10% neutral formalin to fix the tissues for histopathology and subjected to staining with Hematoxylin and Eosin (H&E) staining on lung slices measuring 5 µm to visualize pathology in the lungs of vaccinated mice versus non-vaccinated mice.

### 2.5. Enzyme-Linked Immunosorbent Assay (ELISA)

Total IgG in mice sera and associated binding to WT IAV HAs were assessed using in Immulon 4HBX 96-well flat bottom plates (Thermo Fisher Scientific, Waltham, MA, USA). A total of 100 μL of WT Bris/18 or Sing/16 rHAs, at 1 μg/mL in carbonate coating buffer (pH 9.4) solution was added to the well of each plate and incubated at 4 °C overnight in a humidified chamber. After the incubation, the coating solutions were removed and the wells were blocked for 1.5 h at 37 °C with 200 μL of blocking buffer (BB) containing 4% FBS + 0.05% Tween. The serum samples were made at a 1:100 ratio followed by serially diluting (1:3) from an initial 1:500 dilution. After blocking, 100 μL of diluted sera was added to the WT Bris/18 or Sing/16 coated plates and incubated for 1.5 h at 37 °C. After the incubation, the plates were thoroughly laved, one hundred microliters of secondary goat anti-mouse IgG HRP (Southern Biotech, Birmingham, AL, USA) antibody diluted 1:4000 in blocking buffer was added to each well, followed by statically incubating the plates for 1.5 h at 37 °C. Subsequently, the antibody solution was discarded, each well was thoroughly laved and received one hundred microliters of 1X ABTS (VWR Corporation, Radnor, PA, USA) with static incubation for thirteen minutes at 37 °C. Succeeding incubation, fifty microliters of one percent SDS was added to halt the reaction of each well. IgG titers were measured as the optical density (O.D.) of each sample at 414 nm with a spectrophotometer (PowerWave XS, BioTek, Santa Clara, CA, USA) utilizing the Gen05 software (version 3.14, https://www.agilent.com/en/support/biotek-software-releases (accessed on 1 August 2023)). IgG was compared to positive and negative controls within each plate. To determine IgA and the different Ig isotypes against WT rHAs, the samples were initially handled as before with subsequent static incubated with secondary goat anti-mouse IgA, IgG1, IgG2a, or IgG2b antibodies and the O.D. measured as before. For these sets, sera were made at 1:100 ratio and serially diluted (1:3) from a starting 1:500 dilution for isotype determination, or prepared at 1:10 ratio for IgA and serially diluted (1:2) from the starting 1:10.

### 2.6. Hemagglutination Inhibition Assay (HAI)

Neutralizing anti-HA antibodies are able to block viruses from agglutinating activity red blood cells (RBC). To determine this activity in vaccinated mice sera, the HAI assay was employed. The protocol followed the manual for laboratory diagnosis and virological surveillance of influenza published by the World Health Organization (WHO) [34]. HAI activity was tested against the H1N1 viruses shown herein: TX/36/91, Bris/02/07, Cal/07/09, Bris/02/18, and H3N2 viruses TX/50/12, SW/9715293/13, HK/4801/14, Sing/IFNIMH/16, KS/14/17, SW/8060/17, HK/2671/19, and SA/34/19. Specific details on how the HAI assay was completed has been described [35]. Concisely, sera from individual mice were mixed with receptor-destroying enzyme (RDE) (Denka Seiken, Co., Tokyo, Japan) and diluted to 1/10th by reconstituting one hundred microliters of serum with three volumes of RDE in 1X PBS. The reconstituted samples were then incubated at 37 °C overnight. After overnight incubation, the treated sera were heat inactivated in a water bath at 56 °C for forty-five minutes, allowed to cool to RT, followed by adding six volumes of 1X PBS. Then, 96-well plates (v-bottoms) containing 25 μL of PBS were used to serially diluted the RDE-treated sera samples, in duplicates. Solutions with a ratio of 1:8 of each virus to be tested were added to the plates with serially diluted sera samples and incubated at RT for twenty minutes for the H1N1 influenza viruses or thirty minutes for the H3N2 influenza viruses. After the incubation, H1N1 samples received 0.8% turkey RBC (TRBCs) and H3N2 samples received 0.8% guinea RBC (GPRBCs) and were manually agitated, followed by incubated at RT for thirty minutes. Subsequently, the reciprocal dilutions of the last wells that ran (no agglutination) were reported as the titers. A 1:40 titer was the measurement of seroprotection as per the European Medicines Agency Guidelines on Influenza Vaccines [36].

### 2.7. FluoroSpot Assay

On day 61, spleens were collected (n = 3) per study arm. Following homogenization, splenocytes were washed twice with RPMI 1640 BCM Medium (Gibco^TM^, Grand Island, NY, USA) and centrifuged at 400× *g* for 10 min, at 4 °C. After the second wash, the BCM media was discarded and the cell pellets were resuspended in 4 mL of 90% FBS/10%DMSO freezing media and aliquoted into cryotubes for storage at −80 °C overnight, and then transferred to LN_2_ for future use. To assess antigen-specific antibody secreting cells (ASCs), the Four Color Immunospot^®^ kit (CTL, Shaker Heights, OH, USA) was employed. Following the manufacturer’s instructions, in vivo pre-stimulated splenocytes were washed with BCM and filtered through 70 μm MACS^®^ SmartStrainers (Miltenyi Biotec, San Diego, CA, USA) to removed debris. In v-bottom, 96-well plates, splenocytes were serially diluted three-fold, in duplicates, starting at 3 × 10^5^ live cells per well and transferred to pre-treated (with 70% ethanol) PVDF plates coated with anti-Igκ/λ capture antibody, Y2, NG2, or BSA at 25 μg/mL in Diluent A, provided in the Immunospot kit. Following the addition of splenocytes to the wells, the plates were incubated in a humidified chamber with 5% CO_2_ for 16–18 h at 37 °C, as per the manufacturer’s instructions. Following incubation, the ASCs were removed and the plates were washed twice with 1X PBS. An anti-mouse detection solution containing IgG1/IgG2a/IgG2b was prepared and added to each well, followed by a 2 h incubation in the dark at room temperature (RT). After the incubation period, the plates were washed twice with PBS-T, followed by the addition of a prepared tertiary solution and incubated for 1 h in the dark at RT. After incubation, the plates were decanted, washed twice with distilled water, and allowed to dry overnight in a dark, running laminar flow hood, face down on paper. FluoroSpot detection was achieved upon reading the plates on the ImmunoSpot^®^ S6 Ultimate Analyzer (ImmunoSpot by C.T.L., Shaker Heights, OH, USA). Spot-forming-units (SFUs) were enumerated using the Basic Count mode of the CTL ImmunoSpot SC Studio (Version 1.6.2, Shaker Heights, OH, USA).

### 2.8. Statistical Analysis

The weight loss and clinical scores were statistically evaluated using two-way ANOVA by Prism 9 software (GraphPad Software, Inc., San Diego, CA, USA, version 9.4.0, https://www.graphpad.com (accessed on 1 August 2023). *p*-value less than 0.05 was defined as statistically significant (*p* < 0.05 *, *p* < 0.01 **, *p* < 0.001 ***, *p* < 0.0001 ****). Each line represents an n = 15 per group and is conveyed as the average +/− standard error of the mean (SEM).

For ELISA analysis, IgG, IgG1, IgG2a, and IgG2b titers were statistically evaluated using one-way ANOVA by Prism 9 software (GraphPad Software, Inc., San Diego, CA, USA, version 9.4.0, https://www.graphpad.com (accessed 1 August 2023)). *p*-value less than 0.05 was defined as statistically significant (*p* < 0.05 *, *p* < 0.01 **, *p* < 0.001 ***, *p* < 0.0001 ****). Each bar corresponds to 15 individual mice according to vaccine regimens and are conveyed as the average +/− standard error of the mean (SEM).

The HAI titers were statistically evaluated using one-way ANOVA by Prism 9 software (GraphPad Software, Inc., San Diego, CA, USA, version 9.4.0, https://www.graphpad.com (accessed 1 August 2023)). *p*-value less than 0.05 was defined as statistically significant (*p* < 0.05 *, *p* < 0.01 **, *p* < 0.001 ***, *p* < 0.0001 ****). Each column represents 15 individual mice pertaining to each vaccine regimen and are conveyed as the average +/− standard error of the mean (SEM).

For FluoroSpots analysis, ASC were statistically evaluated using one-way ANOVA by Prism 9 software (GraphPad Software, Inc., San Diego, CA, USA, version 9.4.0, https://www.graphpad.com (accessed 1 August 2023)). *p*-value less than 0.05 was defined as statistically significant (*p* < 0.05 *, *p* < 0.01 **, *p* < 0.001 ***, *p* < 0.0001 ****). Represented is an n = 3 of mice per the indicated vaccine regimens and expressed as the average +/− standard error of the mean (SEM).

## 3. Results

### 3.1. Advax-SM™-Adjuvanted COBRA HA Vaccines Protect Mice against Influenza Virus Challenge

DBA/2J mice were divided into four study arms and vaccinated intramuscularly following prime–boost–boost regimens (Figure 1a,b). Control mice immunized with Advax-SM™ adjuvant alone lost an average ~20% of their original body weight by day 3 post-infection (Figure 2a), with a mean clinical score of 3 (Figure 2b), and all succumbed to infection by day 3 post-infection (Figure 2c). All of the mice vaccinated with COBRA HA proteins formulated with either the Advax-SM™ or comparator SE adjuvant showed maximum weight loss of <5% of their original body weight at day 2 post-infection and returned to their original weight by day 4, with minimal sickness scores particularly in the Advax-SM™ group and no mortality (Figure 2a–c). Mice vaccinated with COBRA HA lost ~11% of their original weight loss by day 3 post-infection with the survivors in this group slowly returning to their original body weight by day 8 post-infection (Figure 2a) with a mean clinical score of 2.08 and only 33% survival (Figure 2c).

Three mice in each group were sacrificed at day 3 post-infection for assessment of lung virus titers and lung pathology. Control Advax-SM™-alone vaccinated mice had high viral lung titers as did the mice that received COBRA HA alone without adjuvant (~1 × 10^6^ pfu/g of tissue) (Figure 3). In contrast, none of the mice vaccinated with the COBRA HA plus Advax-SM™ had detectable lung virus, and only one of three of the COBRA HA plus SE adjuvant immunized mice had detectable, albeit low, viral lung titers.

### 3.2. Advax-SM™ Protects Mice from Developing Lung Pathology following Influenza Infection

Mice vaccinated with COBRA HA proteins plus Advax-SM™ were protected from lung injury with no inflammation (0%) observed in the upper or lower lungs following challenge with Bris/18 (Figure 4c), which was similar to the normal lungs of unchallenged mice (Figure 4f), whereas mice vaccinated with COBRA HA alone (Figure 4a) or COBRA HA plus SE adjuvant (Figure 4b) had evidence of lung inflammation (~50–60%), similar to the inflammation (~60–70%) seen in Mock plus Advax-SM™ and Mock plus saline immunized mice (Figure 4d).

### 3.3. Advax-SM™ Adjuvant Enhances Serum Anti-Influenza IgG, IgA, and Subclass Switching following Vaccination

To assess the anti-influenza IgG and subclasses elicited by each vaccine formulation, sera collected from mice following the last vaccine boost and were assessed for total anti-influenza IgG and IgG isotypes (Figure 5). Mice vaccinated with COBRA plus Advax-SM™ had significantly higher total IgG titers against Bris/18 and Sing/16 rHAs compared to mice vaccinated with COBRA HA plus SE adjuvant or COBRA HA alone (Figure 5a). Mice vaccinated with COBRA HA plus SE adjuvant had a predominance of IgG1 with only a small amount of IgG2a and IgG2b (Figure 5c,d). In contrast, mice vaccinated with the same COBRA HA with Advax-SM™ exhibited a balanced mix of IgG1, IgG2a, and IgG2b against Bris/18 and Sing/16 rHAs (Figure 5c,d). COBRA-only vaccinated mice had significantly less IgG1 with little to no IgG2a or IgG2b (Figure 5c,d) against either WT rHA. Additionally, Mock + Advax-SM™ vaccinated mice had no detectable serum antibodies. Only mice immunized with COBRA HA plus Advax-SM™ had measurable serum anti-influenza IgA (Figure 5b).

### 3.4. Advax-SM™ Adjuvant Enhances Serum Hemagglutination-Inhibition (HAI) Titers in Vaccinated Mice

Vaccinated mice had sera capable of blocking panels of H1N1 and H3N2 influenza viruses from agglutinating RBCs. Mice vaccinated with Advax-SM™ adjuvanted COBRA HA antigens had elevated serum HAI titers against the H1N1 influenza viruses. However, these levels were not statistically different to mice vaccinated with SE adjuvanted COBRA HA antigens (Figure 6). In contrast, only 40–50% of the mice vaccinated with the COBRA HA vaccine alone had HAI responses ≥1:40 against two influenza viruses pertaining to the H1N1 subtype (Figure 6). Mice immunized with COBRA HA plus Advax-SM™ had elevated HAI responses against seven of the eight influenza viruses pertaining to the H3N2 subtype panel (Figure 7c). Comparable findings were perceived in mice immunized with COBRA HA plus SE adjuvant (Figure 7b). There were low to no HAI responses in mice immunized with unadjuvanted COBRA HA vaccines (Figure 7a). None of the mice, regardless of vaccine used, developed serum HAI titers against H1N1 TX/91 or Bris/07.

### 3.5. Advax-SM™ Adjuvant Drives Production of IgG1 and IgG2a Antibody-Secreting Cells

IgG1 and IgG2a antibody-secreting cells (ASC) were identified in the spleens of mice vaccinated with COBRA HA proteins plus Advax-SM™. Sera of these mice had influenza-specific IgG1, IgG2a, and IgG2b antibodies that bound the Y2 and NG2 HA antigens (Figure 8a–c). ASC from mice vaccinated with COBRA HA proteins plus SE adjuvant secreted primarily IgG1 with little or no IgG2a (significantly lower than mice vaccinated with COBRA HA proteins plus Advax-SM™) or IgG2b (Figure 8a–c). Mice vaccinated with unadjuvanted COBRA HA or adjuvant-alone control mice had little to no influenza-specific ASCs (Figure 8a–c).

## 4. Discussion

The first influenza vaccines were developed in the 1930s and 1940s for the U.S. military using embryonated chicken eggs, which is a technique still used to make commercial vaccine today [37,38,39,40,41]. Other vaccine platforms have been approved for commercial seasonal influenza vaccines, including live attenuated virus (LAIV) vaccines and recombinant HA protein vaccines [10,42,43,44]. The first adjuvanted influenza vaccine using MF59, a squalene oil-in-water adjuvant, was approved in 1997 and now is used in 38 countries for people 65 years and older [45,46,47]. Adding this adjuvant to the influenza vaccine broadens the immune responses and allows for increased protection against drift strains of influenza viruses [48,49,50]. Various new adjuvants may have potential to enhance the effectiveness of influenza vaccines [51,52]. A key aim in the influenza field is to identify vaccine formulations that stimulate effective, broadly reactive, and long-lasting protective anti-influenza immune responses in all age groups. Newer adjuvants include toll-like receptor (TLR) agonists such as GLA (glucopyranosyl lipid A), a TLR4 agonist, or CpG oligodeoxynucleotides, which are TLR9 agonists, as well as saponin adjuvants (e.g., Iscomatrix or Matrix M) that act via inflammasome activation [53,54,55]. Falling into its own unique category, Advax-SM™ is a new particulate polysaccharide adjuvant plus CpG55.2, a TLR9 agonist. In this study, COBRA HA antigens were adjuvanted with either Advax-SM™ or a control SE adjuvant to assess potential differences in the action of these adjuvants. Advax-SM™ adjuvant appears to work via enhanced non-inflammatory recruitment of immune cells to the site of vaccination, leading to enhanced antigen presentation to influenza-specific B cells and CD4 and CD8 T cells [25,56]. SE adjuvant works by inducing local inflammation that recruits immune cells and also helps to form an antigen depot [57,58,59]. Unlike some adjuvants that can be highly reactogenic, natural polysaccharides, such as delta inulin, are non-reactogenic, biodegradable, and have a strong safety profile and also have good long-term, room temperature stability [60]. Advax-SM™ adjuvant provides antigen dose-sparing and induces long-lasting T-cell and humoral immune responses, while exhibiting low local or systemic reactogenicity as compared to oil emulsion or saponin adjuvants [25].

In this study, mice vaccinated intramuscularly with the COBRA HA proteins only without adjuvant lost more than 20% body weight following influenza virus infection, exhibited severe lung inflammation, and had high mortality. These mice had high viral lung titers that were similar to adjuvant-alone vaccinated mice, which also rapidly died after infection. The same COBRA HA when formulated with Advax-SM™ or the oil-in-water adjuvant, SE adjuvant (similar to MF59), induced high serum HAI activity that translated to complete survival with minimal weight loss, and undetectable lung virus, apart from one of three animals in the COBRA HA plus SE adjuvant group. When comparing weight loss and clinical score of mice vaccinated with COBRA HA alone to COBRA HA plus SE adjuvant or Advax-SM^TM^, there was significant weight loss between days 3–4 post-infection and significant clinical score between days 2–6 post-infection with both adjuvanted vaccines. Mice vaccinated with COBRA HA proteins plus Advax-SM™ had ~4-fold higher HAI titers against H1N1 and most H3N2 influenza viruses when compared to mice vaccinated with COBRA HA and SE adjuvant. Notably, mice vaccinated with COBRA HA proteins plus Advax-SM™ exhibited no inflammation in their lungs, whereas those vaccinated with COBRA HA + SE adjuvant still exhibited significant lung inflammation similar to mice immunized with COBRA HA alone. This suggests that the two adjuvants might be working via different immune pathways. Mice vaccinated with SE adjuvant predominately generated IgG1 ASC that is indicative of a dominant T helper type 2 (Th2) CD4+ T cell response and therefore resulted in a predominately IgG1 response against Bris/18 and Sing/16 rHAs, with low titers of IgG2a and IgG2b and no IgA. In contrast, mice vaccinated with COBRA HA plus Advax-SM™ had a mixed T helper response with increased IgG1 and IgG2a (IgG2a significantly higher than SE adjuvanted vaccines) ASC and higher serum IgG2a, IgG2b, and IgA (IgA not significant to mice vaccinated with SE adjuvanted vaccines) antibody titers than the other study arms. The induction of T-bet and STAT-4 signaling pathways by IFN-γ activation skews immune responses towards a T helper type 1 (Th1) response, whereas STAT-6 and GATA-3 following IL-4 and IL-2 activation skews towards a Th2 responses [61,62,63,64]. Hence, IFN-γ stimulates IgG isotype switching to IgG2a, a Th1 phenotype, whereas IL-4 induces IgG1 indicative of the Th2 phenotype [65,66,67]. While the mechanism of action of Advax-SM™ is not fully known, Advax-SM™ may function through enhanced antigen uptake by recruiting larger numbers of antigen-presenting cells (APC), altered antigen processing, and stimulation of different cytokine profiles than SE adjuvant oil-in-water emulsions, leading to enhanced B- and T-cell activation [25]. The induction of mixed T helper responses and a broader range of IgG isotypes and IgA by the Advax-SM™-adjuvanted COBRA HA may help explain the ability of this vaccine to reduce morbidity and mortality and, in particular, prevent lung inflammation in response to influenza virus infection compared to the SE adjuvant squalene emulsion adjuvant.

Delta inulin also activates the alternative complement pathway (ACP) resulting in the production of C3d; a degradation of the third component of complement (C3) [68]. C3d covalently couples to antigens, enhancing their immunogenicity [69]. C3d binds to the surface complement receptor 2 (CR2) of follicular dendritic cells (FDC) and B and T-cells [69]. Via FDCs, C3d stimulates antigen presentation and interacts with CR2 inducing it to co-localize with molecules such as CD19, TAPA (CD81), and Lew 13 [70]. CD19 triggers a signaling cascade that results in cell activation and proliferation that in conjunction with C3d-CR2 ligation and surface immunoglobulin (sIg) binding by the vaccine antigen, activates two signaling pathways that cross-talk and synergize to activate B cells and subsequent Ig class switching and elicitation of high titer, antigen-specific antibodies [68,69].

## 5. Conclusions

Mice vaccinated with COBRA HA plus Advax-SM™ had increased serum IgG1, IgG2a, IgG2b, and IgG1 and IgG2a ASCs directed against H1 and H3 HA. Hence, the ability of Advax-SM™ adjuvant to activate complement may contribute, at least in part, to its beneficial adjuvant effects on the COBRA HA vaccines.

Mice immunized with COBRA HA plus Advax-SM™ adjuvant had both Th1 and Th2 anti-influenza immune responses. Th1 responses and, in particular, cytotoxic CD8 T cells play a key role in clearance of IAV infection [71,72,73]. The ability of Advax-SM™ to induce strong Th1 responses may therefore play a key role in virus clearance. A simultaneous Th2 response induced by Advax-SM™ adjuvant may be beneficial to maximize HAI antibody levels to neutralize the virus while also helping to counteract any excessive virus-induced inflammation [74].

Overall, these findings demonstrate that Advax-SM™, as a balanced Th1 and Th2 adjuvant, was able to enhance the humoral and cellular immunogenicity and protection provided by the next-generation COBRA HA vaccines as compared to COBRA HA alone or formulated with SE adjuvant squalene emulsion adjuvant. The combination of COBRA HA with Advax-SM™ adjuvant therefore represents a promising strategy for enhancing influenza protection in immunosuppressed and high-risk human populations.

## Figures and Tables

**Figure 1 vaccines-12-00455-f001:**
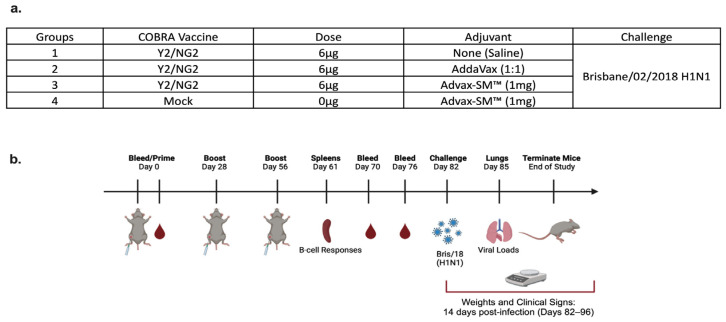
(**a**) Mice were randomly divided into four groups, with n = 18 mice in each group, and immunized intramuscularly (IM) with 3 μg of each COBRA HA proteins, Y2, and NG2 (total 6 μg), alone or formulated with AddaVax SE adjuvant at a 1:1 ratio or 1 mg Advax-SM™. As controls, a group of mice were immunized with 1 mg Advax-SM™ adjuvant only without antigen. (**b**) Schematic of study timeline. Mice were bled and prime-vaccinated on day 0 and boosted on days 28 and 56. On day 61, three mice from each group were sacrificed and spleens harvested for B cell FluoroSpot assays. Bleeds on remaining mice were performed on days 70 and 76, and an intranasal virus challenge was performed on day 82. Three days post-infection (day 85), lungs were harvested from three mice per group for lung viral titers and pathology, and the remaining mice were monitored for clinical illness and mortality for 14 days post infection.

**Figure 2 vaccines-12-00455-f002:**
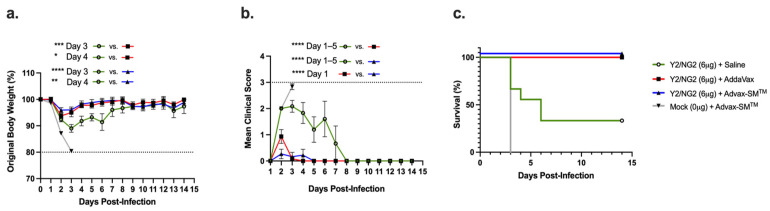
Mice were all challenged intranasally (IN) with the H1N1 strain, A/Brisbane/02/2018 (8 × 10^6^ PFU/50 μL), and observed for 14 days post-infection. (**a**) Percent of original body weight loss, (**b**) clinical scores, and (**c**) percent survival. The dotted line in (**a**) represents the 20% weight loss endpoint cutoff. The dotted line in (**b**) represents a mean clinical score of 3. Each line (**a**,**b**) is conveyed as the average +/− standard error of the mean (SEM). *p* < 0.05 *, *p* < 0.01 **, *p* < 0.001 ***, *p* < 0.0001 ****.

**Figure 3 vaccines-12-00455-f003:**
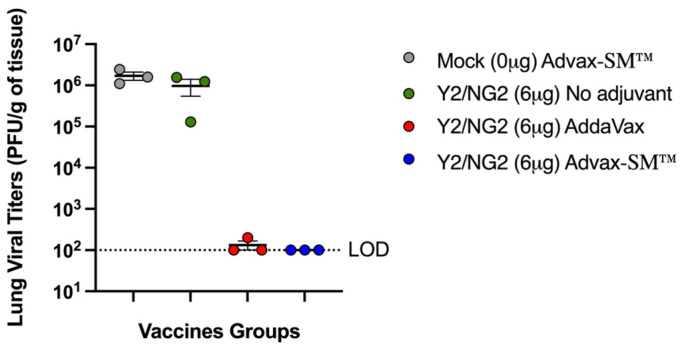
Lung viral titers of mice three days following challenge with A/Brisbane/02/2018. The Y-axis represents the day 3 post-challenge lung viral titers (PFU/g of tissue) and the X-axis represents the vaccine groups. Advax-SM™ alone control (Grey symbols), COBRA HA alone (Green symbols), COBRA HA plus Advax-SM™ (Blue symbols), COBRA HA plus SE adjuvant (Red symbols). The dotted line represents the limit of detection (LOD).

**Figure 4 vaccines-12-00455-f004:**
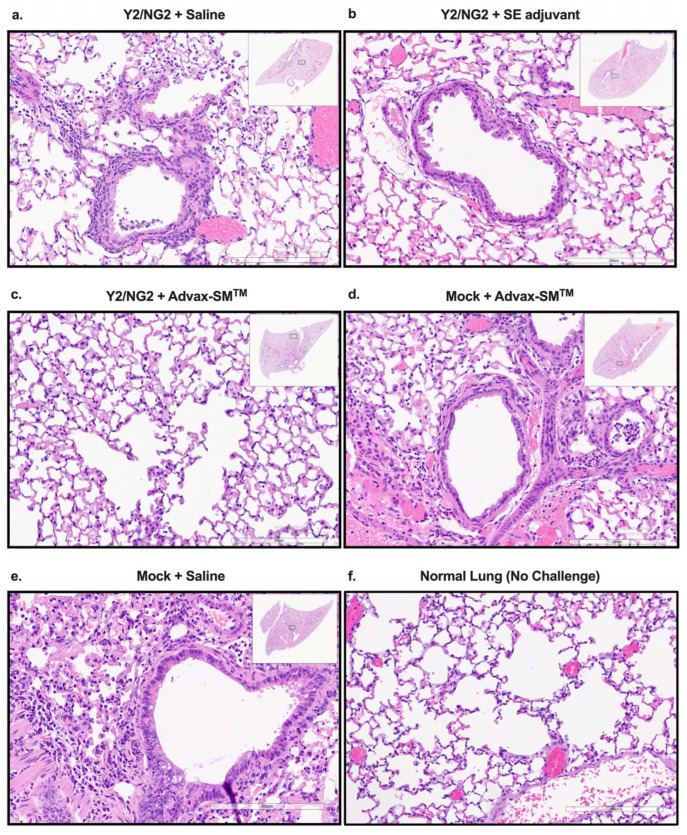
Histopathology of mice lung tissue harvested at three days post-infection with H1N1 Brisbane/02/18 influenza virus. The mice were humanely euthanized and their left lungs were infused with 10% formalin for fixing the tissue. Hematoxylin and Eosin (H&E) staining on lung slices measuring 5 µm was used to visualize pathology in the lungs. The lungs are represented at 20× magnification—the lower bar represents 200 um scale. COBRA HA plus (**a**) saline, (**b**) SE adjuvant, or (**c**) Advax-SM^TM^. Saline plus (**d**) Advax-SM^TM^, Mock plus (**e**) saline, or (**f**) uninfected.

**Figure 5 vaccines-12-00455-f005:**
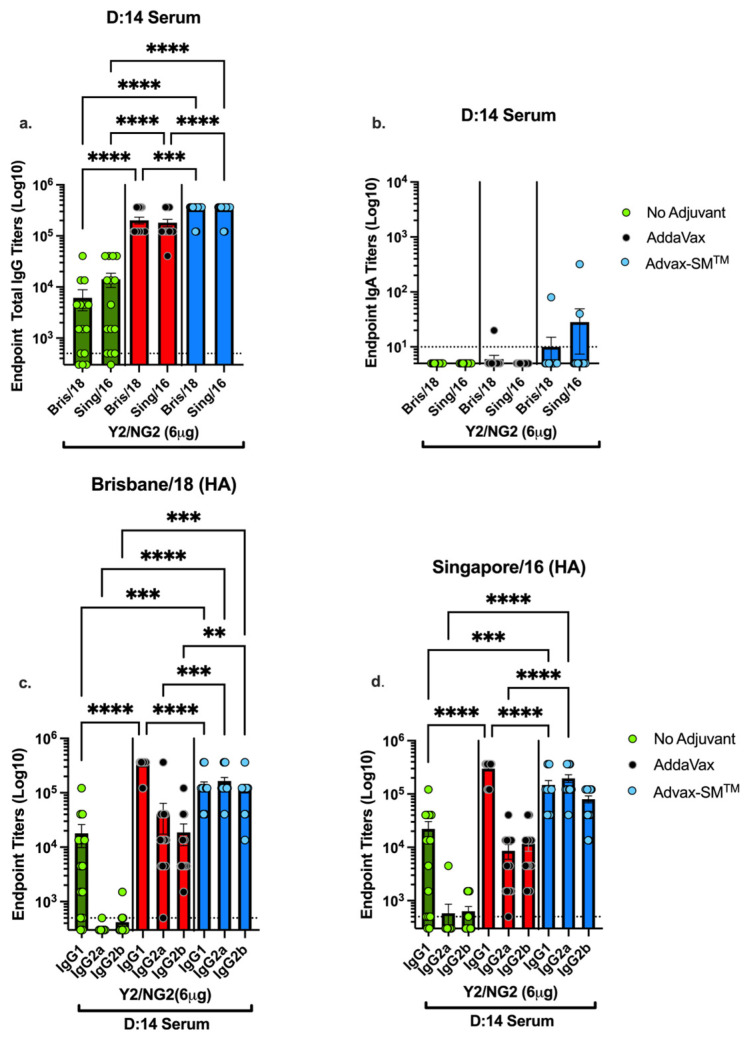
Serum anti-influenza total IgG, IgA and IgG1, IgG2a, and IgG2b in vaccinated mice. (**a**) Serum anti-influenza total IgG against Bris/02/18 and Sing/16 rHA, (**b**) serum IgA against WT Bris/18 and Sing/16 rHA, (**c**) serum IgG isotype titers against WT Bris/18, and (**d**) serum IgG isotype titers against Sing/16 rHA. Represented on the Y-axis are the endpoint titers. Represented on the X-axis are the WT rHAs (**a**,**b**) or the Ig isotypes (**c**,**d**). Each bar corresponds to 15 individual mice according to vaccine regimens and are conveyed as the average +/− standard error of the mean (SEM). *p* < 0.01 **, *p* < 0.001 ***, *p* < 0.0001 ****.

**Figure 6 vaccines-12-00455-f006:**
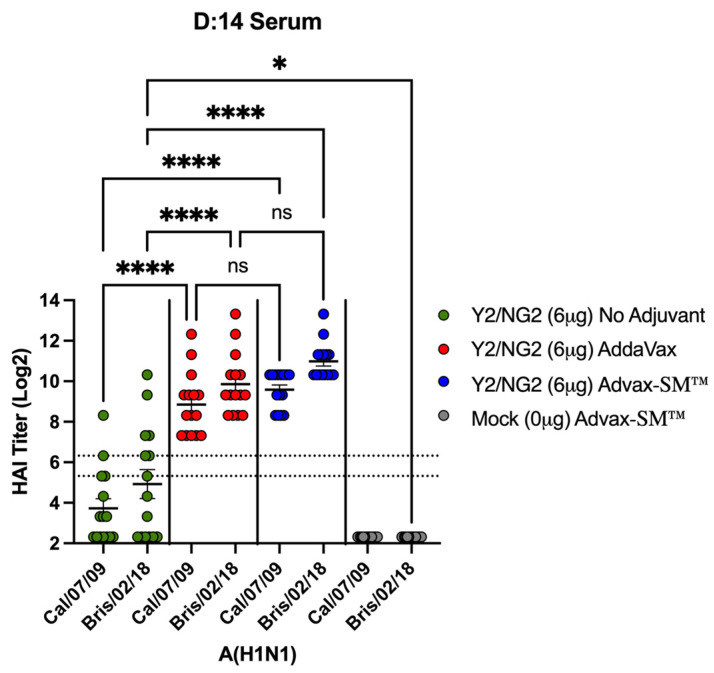
Hemagglutination inhibition activity in mice vaccinated with COBRA HA vaccines (Y2 and NG2). Sera collected from immunized mice at days 70 and 76 were pooled (3 weeks post the third vaccine dose) and tested for HAI activity against two H1N1 viruses, Cal/09 and Bris/18. COBRA HA alone (Green symbols), COBRA HA + SE adjuvant (1:1) (Red symbols), COBRA HA + Advax-SM™ (Blue symbols), or Mock + Advax-SM™ (Grey symbols). Represented on the Y-axis are the HAI titers on a log 2 scale. Represented on the X-axis is the panel of H1N1 influenza viruses. The top dotted line indicates a 1:80 HAI titer and the bottom dotted line indicates a 1:40 HAI titer. Each column represents 15 individual mice pertaining to each vaccine regimen and are conveyed as the average +/− standard error of the mean (SEM). *p* < 0.05 *, *p* < 0.0001 ****, *p* < 0.9999 not significant (ns).

**Figure 7 vaccines-12-00455-f007:**
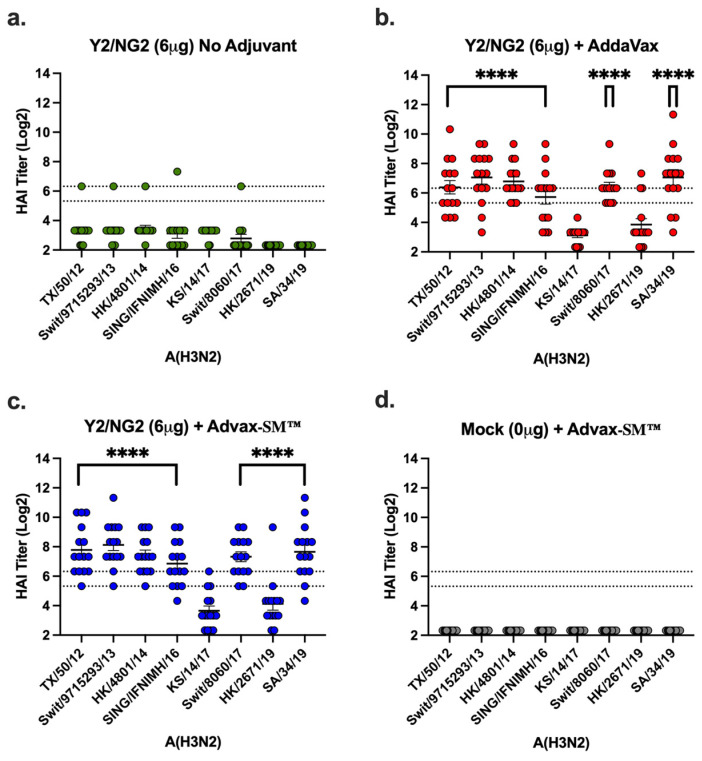
Hemagglutinin inhibition responses in mice immunized with COBRA HA antigens (Y2 and NG2). (**a**–**c**) Sera collected from immunized mice at days 70 and 76 were pooled (3 weeks post the third vaccination) and tested against two H3N2 influenza viruses. (**a**) Advax-SM™ alone controls, (**b**) COBRA HA + SE adjuvant (1:1), (**c**) COBRA HA + Advax-SM™, or (**d**) Mock + Advax-SM™. Represented on the Y-axis are the HAI titers on a log 2 scale. Represented on the X-axis is the panel of H3N2 influenza viruses. The top dotted line indicates a 1:80 HAI titer and the bottom dotted line indicates a 1:40 HAI titer. Statistical differences shown are of group (**a**) compared to group (**b**) or (**c**). Each column represents 15 individual mice pertaining to each vaccine regimen and are conveyed as the average +/− standard error of the mean (SEM). *p* < 0.0001 ****.

**Figure 8 vaccines-12-00455-f008:**
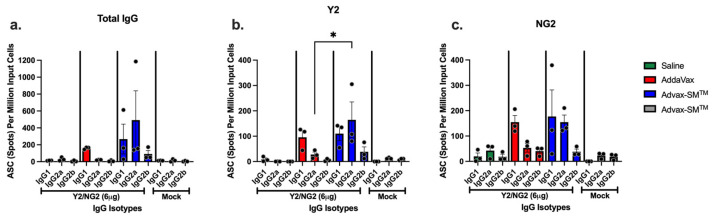
Quantification of total IgG and antigen-specific IgG1, IgG2a, and IgG2b isotypes from antibody secreting cells (ASCs). Spleens from DBA/2J female mice (6–8 weeks old) were harvested five days following the third vaccination with COBRA HA+ SE adjuvant (Red), COBRA HA+ Advax-SM™ (Blue), Mock + Advax-SM™ (Grey), or COBRA HA alone (Green). Plates coated with anti-Igκ/λ (**a**), Y2 (**b**), or NG2 (**c**) were used to detect IgG1, IgG2a, or IgG2b ASCs. The Y-axis represents the number of ASCs (spots) per million input cells. The Ig isotypes from immunized mice are represented on the X-axis. Each column represented 3 individual mice per the indicated vaccine regimens and expressed as the average +/− standard error of the mean (SEM). *p* < 0.05 *.

**Table 1 vaccines-12-00455-t001:** Multiple alignment for all full-length H1N1 HA sequences included in this study.

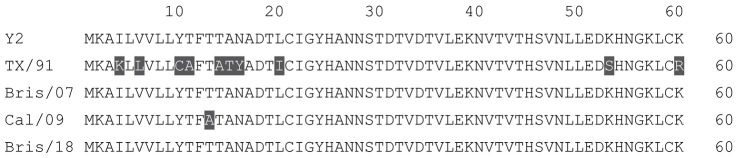
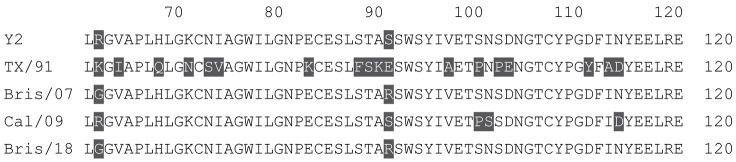
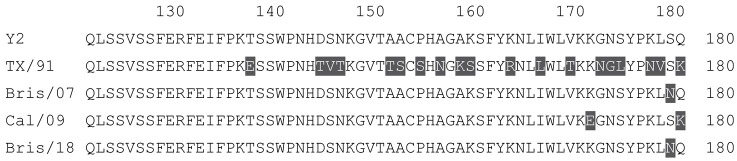
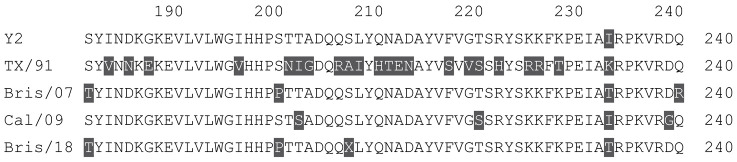
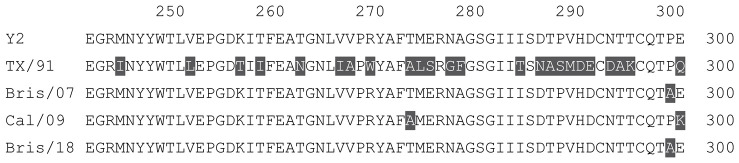
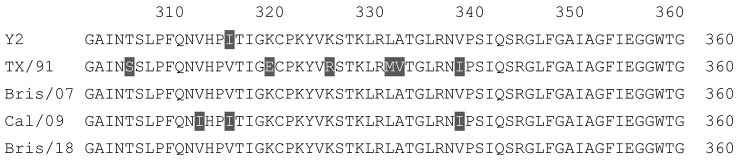
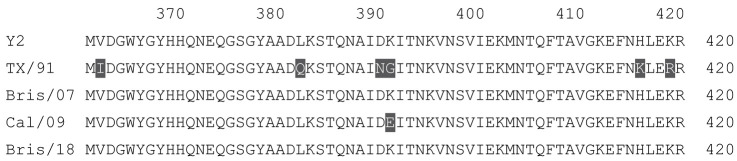
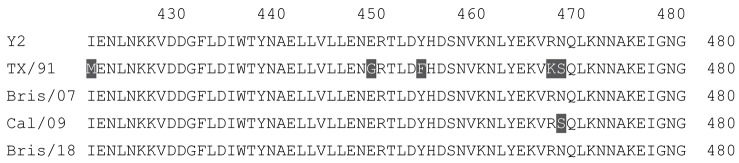
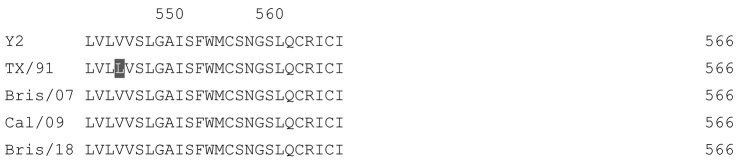

The full-length Y2 COBRA HA amino acid sequence with enumeration listed above. Four wild-type H1 HA sequences are listed with viruses isolated in 1991, 2007, 2009, and 2018. Any amino acid position that has a different residue at the position than the Y2 HA sequence is highlighted in grey.

**Table 2 vaccines-12-00455-t002:** Multiple alignment for all full-length H3N2 HA sequences included in this study.

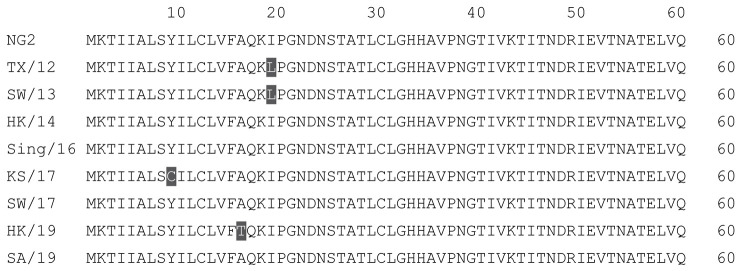
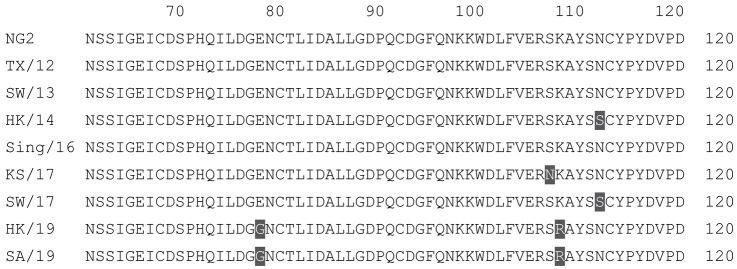
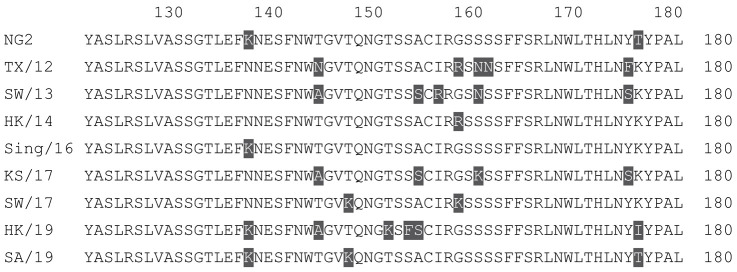
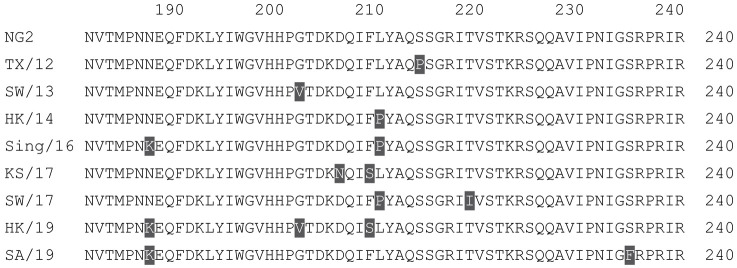
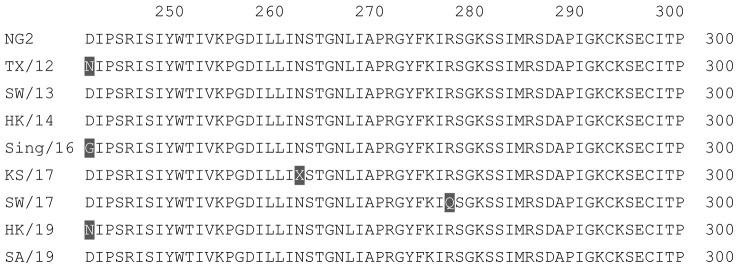
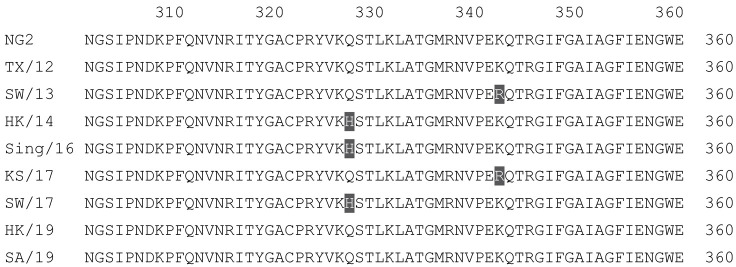
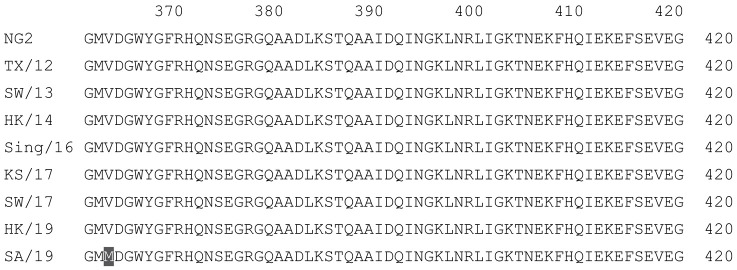
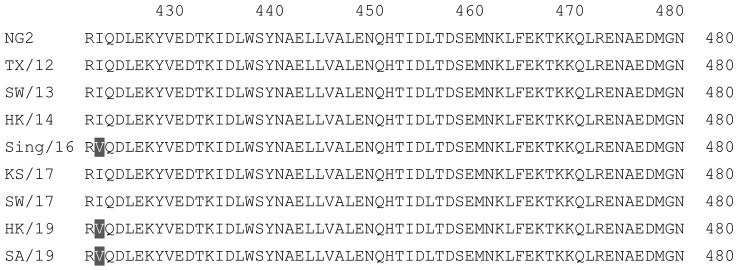
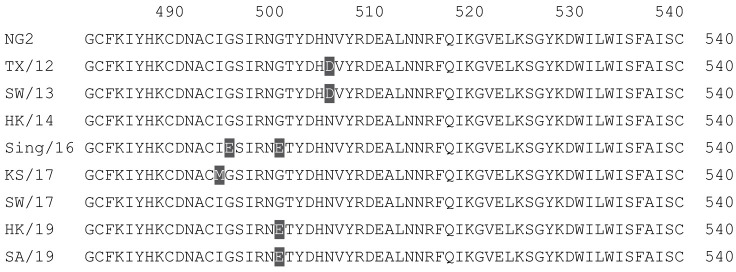
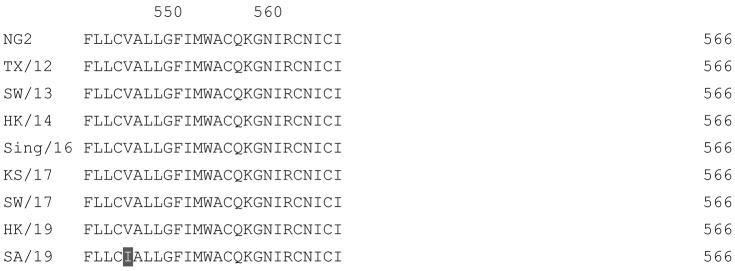

The full-length NG2 COBRA HA amino acid sequence with enumeration listed above. Eight wild-type H3 HA sequences are listed with viruses isolated in 2012, 2013, 2014, 2016, 2017, and 2019. Any amino acid position that has a different residue at the position than the NG2 HA sequence is highlighted in grey.

## Data Availability

The data presented in this study may be made available upon written request to the corresponding author.

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
