# Peer review of "Advax-SM™-Adjuvanted COBRA (H1/H3) Hemagglutinin Influenza Vaccines"

_vaccines, 2024, doi:10.3390/vaccines12050455_

Round 1

Reviewer 1 Report

Comments and Suggestions for Authors

Influenza vaccinations play a crucial role in public health by providing immunity against various strains of the virus. Previously, the authors have developed vaccines using computationally optimized influenza antigens to elicit antibodies covering multiple strains. The primary objective of this paper was to assess the efficacy of the Advax-SMTM adjuvant in enhancing the immune response generated by these computationally designed antigens. Utilizing a mouse model, the research evaluated various immunological parameters, including body weight, survival rates, lung viral titers, and serum antibody levels, to determine the adjuvant's effectiveness. The findings suggest that Advax-SMTM significantly boosts the immune response, underscoring the research's value and its contributions to the field. Nonetheless, for this manuscript to be considered for publication, certain methodological and presentation aspects should be improved:

The results imply a stark contrast in vaccine efficacy with and without the adjuvant, raising questions about the intrinsic effectiveness of the COBRA Y2/NG2 vaccine itself. The authors should clarify whether the vaccine's design inherently lacks broad-spectrum coverage. This is my most fundamental question regarding the results as I would think the vaccine itself should be reasonably effective already

In Figure 2b, the criteria for determining the clinical score are not delineated within the methods section. A detailed explanation of this scoring system should be included in the methods section.

Figure 4 has a mock+saline group, but this group is not mentioned in Figure 1a. The origin and rationale behind this group's inclusion should be clarified.

Figures 5, 6, and 7 omit the mock+adjuvant group. Incorporating this data could provide a comprehensive negative control for comparison.

The data points in Figures 5, 6, and 7 appear uniformly spaced, which may result from data rounding or the assay's characteristics. The authors should provide raw data files or the Prism file would enhance transparency and allow for a more detailed analysis.

While the rationale for selecting Advax-SMTM as an adjuvant is discussed, the justification for using the SE adjuvant remains unaddressed in the introduction section.

The labeling of positions in Tables 1 and 2 appears misaligned (10, 20, 30, 40, 50, 60 look off axis from the actual position).

Author Response

Reviewer 1

The results imply a stark contrast in vaccine efficacy with and without the adjuvant, raising questions about the intrinsic effectiveness of the COBRA Y2/NG2 vaccine itself. The authors should clarify whether the vaccine's design inherently lacks broad-spectrum coverage. This is my most fundamental question regarding the results as I would think the vaccine itself should be reasonably effective already

Response:  Thank you for bringing this up. This is a valid point. In general, adjuvants are essential to enhance the immunogenicity of subunit vaccines, such as recombinant HAs [1]. One potential way that adjuvants may do this is in a host is by enhancing the deliverance of the administered vaccines antigens in a timely manner in which the antigen is protected from the host immune responses, thus leading to degradation of the exposed immunogens [2]. HA COBRA vaccines administered alone provide little to no immunogenicity [3] unless adjuvanted with a properly tailored adjuvant, which is one of the important aspects of this study. According to the results shown here, Advax was able to enhance the immunogenicity of these COBRA HA vaccines. This translated to the immunological responses associated with broad-spectrum coverage where mice vaccinated with Advax-adjuvanted COBRA HA vaccines seroconverted against panels of a variety of H1N1 and H3N2 influenza viruses (Figures 6 and 7) as well as inducing antibodies of different isotypes that are specific to different WT influenza virus rHAs (Figure 5). To add to the success of the vaccine design, mice vaccinated with Advax-adjuvanted COBRA HA vaccines were protected from morbidity and mortality following infection with influenza virus (Figures 2, 3, and 4).

In Figure 2b, the criteria for determining the clinical score are not delineated within the methods section. A detailed explanation of this scoring system should be included in the methods section.

Response:  Thank you for pointing this out. A detailed explanation of the scoring system is included in the Material and methods section, lines 140-144.

Figure 4 has a mock+saline group, but this group is not mentioned in Figure 1a. The origin and rationale behind this group's inclusion should be clarified.

Response:  Thank you. This point is valid. However, this mock+saline group is a comparator used for qualitative analysis of harvested lung pathology following infection only (no weight loss or clinical scores) and was done separately under the same conditions, using the same mouse strain and virus/dose of virus. This has also been mentioned in lines 120-121 for clarification.

Figures 5, 6, and 7 omit the mock+adjuvant group. Incorporating this data could provide a comprehensive negative control for comparison.

Response:  Thank you for the suggestion. Figures 6, 7 has been updated to include the mock+adjuvant group and legend modified in lines 367-368 and 378-379. Mock+adjuvant group did not seroconvert and had no antibody secreting cells. This has been specified in lines 336 for clarification.

The data points in Figures 5, 6, and 7 appear uniformly spaced, which may result from data rounding or the assay's characteristics. The authors should provide raw data files or the Prism file would enhance transparency and allow for a more detailed analysis.

Response:  Thanks for the comment. The raw data prism files have been provided with the groups highlight in yellow.

While the rationale for selecting Advax-SMTM as an adjuvant is discussed, the justification for using the SE adjuvant remains unaddressed in the introduction section.

Response:  Thank you for pointing this out. To address this justification in the introduction section, modifications have been made in lines 74-78.

The labeling of positions in Tables 1 and 2 appears misaligned (10, 20, 30, 40, 50, 60 look off axis from the actual position).

Response:  Thank you for pointing this out. The tables 1 and 2 have been aligned according.

Reviewer 2 Report

Comments and Suggestions for Authors

The authors present an innovative study comparing to use of adjuvants with engineered influenza HA proteins. This work is important because finding antigen/adjuvant combinations that optimize broadly protective immune responses to influenza is important for developing more efficacious influenza vaccines. The authors found that COBRA HA plus Advax-SM immunized mice were protected against both morbidity and mortality following viral challenge and, at postmortem, had no detectable lung viral titers or lung inflammation. Overall, the Advax- SM-adjuvanted COBRA HA formulation provided effective protection against drifted H1N1 and H3N2 influenza viruses. The paper is well-written with good experimental designs and executions. The paper will add important information to the vaccine field and will be of interest to a broad audience. The paper is appropriate for publication with minor text corrections:

Line 293.  Figure 1 should be mentioned in the Results section before Figure 2.

Maybe the first sentence should be like “ Mice were immunized in four groups in this study and a prime and two boosts (Fig. 1a-1b).   Control mice….

Line 340.  Should 4a be 5a.

Line 347 should 4c and d be 5c and d.

Line 390 should 6a, 6b, 6c be 7a, 7b, 6c.

Author Response

Reviewer 2

The authors present an innovative study comparing to use of adjuvants with engineered influenza HA proteins. This work is important because finding antigen/adjuvant combinations that optimize broadly protective immune responses to influenza is important for developing more efficacious influenza vaccines. The authors found that COBRA HA plus Advax-SM immunized mice were protected against both morbidity and mortality following viral challenge and, at postmortem, had no detectable lung viral titers or lung inflammation. Overall, the Advax- SM-adjuvanted COBRA HA formulation provided effective protection against drifted H1N1 and H3N2 influenza viruses. The paper is well-written with good experimental designs and executions. The paper will add important information to the vaccine field and will be of interest to a broad audience. The paper is appropriate for publication with minor text corrections:

Line 293.  Figure 1 should be mentioned in the Results section before Figure 2.

Maybe the first sentence should be like “ Mice were immunized in four groups in this study and a prime and two boosts (Fig. 1a-1b).   Control mice….

Response: Thank you for pointing this out. Lines number 278-279 in the results section have been modified appropriately to follow the order of figure 1 before figure 2.

Line 340.  Should 4a be 5a.

Response: You are right and thank you for pointing this out. Line 326 has been modified to the correct figure number 5.

Line 347 should 4c and d be 5c and d.

Response: That is correct. This has also been modified accordingly in lines 334.

Line 390 should 6a, 6b, 6c be 7a, 7b, 6c.

Response: This, that is correct. This has also been modified accordingly in line 378.

Reviewer 3 Report

Comments and Suggestions for Authors

The manuscript “Advax-SM™-adjuvanted COBRA (H1/H3) Hemagglutinin Influenza Vaccines” by Sanchez et al, has used commercial adjuvant Advax-SM and reported the enhancement of serum anti influenza antibodies as well as the HAI titer compared to the control.

The article is well organized and clearly written and I have some minor comments to the manuscript mentioned below.

1.     Line 91 COBRA HAs, in soluble or soluble nature?

2.     Table 1 can be added as supplementary table to concise the pages in the manuscripts.

3.     Line 159 “Reb blood cells s/b change to Red blood cells.

4.     Line 218 WT IAV Has, it s/b HAs

5.     Figure 6 and Fig 7 “The top dotted line indicates a 1:40 HAI titer and the top dotted line indicates a 1:80 HAI titer”. It’s confusing, it s/b top dotted line and bottom dotted line.

Author Response

The manuscript “Advax-SM™-adjuvanted COBRA (H1/H3) Hemagglutinin Influenza Vaccines” by Sanchez et al, has used commercial adjuvant Advax-SM and reported the enhancement of serum anti influenza antibodies as well as the HAI titer compared to the control.

The article is well organized and clearly written and I have some minor comments to the manuscript mentioned below.

  1. Line 91 COBRA HAs, in soluble or soluble nature?

Response: Thank you for pointing this out. The text has been modified for clarity to soluble COBRA HAs in line 93.

  1. Table 1 can be added as supplementary table to concise the pages in the manuscripts.

Response: Thank you for the suggesting. This will be taken into consideration during the submission process.

  1. Line 159 “Reb blood cells s/b change to Red blood cells.

Response: Thanks for noticing this. This text has been modified in line 127.

  1. Line 218 WT IAV Has, it s/b HAs

Response: Thanks for noticing this. This text has been modified in line 193.

  1. Figure 6 and Fig 7 “The top dotted line indicates a 1:40 HAI titer and the top dotted line indicates a 1:80 HAI titer”. It’s confusing, it s/b top dotted line and bottom dotted line.

Response: Thank you for noticing this. The text in legends for figures 6 and 7 have been modified accordingly in lines 369-370 and 381-382.

Reviewer 4 Report

Comments and Suggestions for Authors

This paper describes the experimental results obtained in mice that received COBRA HA vaccines combined with Advax-SM™ as regards qualitative and quantitative immune responses, tissue damage, morbidity and mortality. The topic is extremely captivating and the manuscript is well written. However, some issues about the presentation of the results require urgent attention:

1.      Figure 1 is not understandable enough. Please correct the image in order to include the details reported in the caption.

2.      In “Materials and Methods” section, a paragraph detailing the statistical analysis is missing. Information about the used approaches and software should be removed from the captions of Figures 5 and 6, and described in a specific paragraph.

3.      In Paragraph 3.1 of “Results” section, no statistical analysis was performed. It is impossible to evaluate the significance of the results. Similarly, in Figure 7 there is no trace of statistical analysis of the presented data, and the corresponding text in Paragraph 3.4 does not add any piece of information in this regard. In Paragraph 3.5, no statistically significant results are commented, and from the analysis of Figure 8 (reporting data extrapolated from Paragraph 3.5), it seems that some statistical analysis was performed only for data in panel c.

4.      There is no Institutional Review Board Statement and approval number for this study (that involves animals).

5.      The evaluation of raw data has been impossible, because the file extension is only compatible with GraphPad Prism.

Also, some minor issues were detected:

6.      Please be sure to explain all the acronyms and abbreviations (e.g., SE, stable emulsion).

7.      Correct ug with μg.

Author Response

This paper describes the experimental results obtained in mice that received COBRA HA vaccines combined with Advax-SM™ as regards qualitative and quantitative immune responses, tissue damage, morbidity and mortality. The topic is extremely captivating and the manuscript is well written. However, some issues about the presentation of the results require urgent attention: 

  1. Figure 1 is not understandable enough. Please correct the image in order to include the details reported in the caption.

Response: Thank you for your comment. Figure 1 and the legend associated with it has been modified accordingly for clarification in lines 153-155.

  1. In “Materials and Methods” section, a paragraph detailing the statistical analysis is missing. Information about the used approaches and software should be removed from the captions of Figures 5 and 6, and described in a specific paragraph.

Response: Thank you for the suggestion. The approaches and software used for statistical analysis has been removed from the captions of figures 5 and 6 and added into the material and methods section in lines 217-220 for figure 5 and 243-246, for figure 6.

  1. In Paragraph 3.1 of “Results” section, no statistical analysis was performed. It is impossible to evaluate the significance of the results. Similarly, in Figure 7 there is no trace of statistical analysis of the presented data, and the corresponding text in Paragraph 3.4 does not add any piece of information in this regard. In Paragraph 3.5, no statistically significant results are commented, and from the analysis of Figure 8 (reporting data extrapolated from Paragraph 3.5), it seems that some statistical analysis was performed only for data in panel c.

Response: Thank you for this observation. Statistical analysis was performed for figure 7 and the figure and text in lines 243-246 have been updated accordingly. Additionally, there was some statical analysis done for figure 8b and the software used has been mention in lines 272-275, and the text in 3.5 has been modified in lines 390-391 to mention this.

  1. There is no Institutional Review Board Statement and approval number for this study (that involves animals).

Response: Thank you for pointing this out. An institutional Review Board Statement and approval number for this study had been added in lines 514-517.

  1. The evaluation of raw data has been impossible, because the file extension is only compatible with GraphPad Prism.

Response: Thank you for bringing this up. We would gladly provide the raw data in another format that is compatible upon request.

Also, some minor issues were detected: 

  1. Please be sure to explain all the acronyms and abbreviations (e.g., SE, stable emulsion).

Response: Thank you for the suggesting. SE stands for squalene emulsion and this has been modified in line 15-16 for clarification.

  1. Correct ug withμg. 

Response: Thank you for noticing this. ug has been modified with μg throughout the manuscript and figures accordingly.

Round 2

Reviewer 1 Report

Comments and Suggestions for Authors

The authors have addressed all my concerns. 

Author Response

Thank you

Reviewer 4 Report

Comments and Suggestions for Authors

This reviewer would like to thank the Authors for improving the manuscript according to the proposed suggestion. As previously recommended, this reviewer again suggests that the information about statistical analysis/software is included in a dedicated paragraph, rather than reiterated in Paragraphs 2.3, 2.5, 2.6 and 2.7.

Author Response

Review Report Round 2

This reviewer would like to thank the Authors for improving the manuscript according to the proposed suggestion. As previously recommended, this reviewer again suggests that the information about statistical analysis/software is included in a dedicated paragraph, rather than reiterated in Paragraphs 2.3, 2.5, 2.6 and 2.7.

Response: Thank you for pointing this out. The information about statistical analysis/software has been included in a dedicated paragraph (2.8 in the materials and methods) rather than in the text, as requested, in lines 263-279, and highlighted.